# The Effect of Ca and Mg on the Microstructure and Tribological Properties of YPbSn10 Antifriction Alloy

**DOI:** 10.3390/ma15093289

**Published:** 2022-05-04

**Authors:** Vasile Avram, Ioana Csaki, Ileana Mates, Nicolae Alexandru Stoica, Alina-Maria Stoica, Augustin Semenescu

**Affiliations:** 1Faculty of Materials Science and Engineering, University POLITEHNICA Bucharest, 313 Splaiul Independentei, 060042 Bucharest, Romania; avram.vasile@upb.ro (V.A.); ileana_mariana.mates@upb.ro (I.M.); augustin.semenescu@upb.ro (A.S.); 2Faculty of Mechanical Engineering and Mechatronics, University POLITEHNICA Bucharest, 313 Splaiul Independentei, 060042 Bucharest, Romania; nicolae.stoica@upb.ro (N.A.S.); am.stoica@upb.ro (A.-M.S.); 3Academy of Romanian Scientists, 3 Ilfov, 050044 Bucharest, Romania

**Keywords:** microalloying, friction coefficient, homogeneous microstructure

## Abstract

An alloy YPbSn10 used for antifriction applications was synthetized in a furnace and the structure was improved by a microalloying technique. The elements chosen for microalloying were Ca 2%wt and Mg 2%wt. The microalloying technique proved to have good results in producing alloys with homogeneous composition, with a good distribution of the hard phase. The alloys were produced in a furnace and samples were collected and investigated. The structural properties were investigated using an SEM technique with EDS analyses and XRD to identify the compounds formed during alloying. The tribological properties were investigated to see the improvement obtained in this area. The results revealed a homogeneous composition for both samples, alloyed with Ca or with Mg, and the friction coefficient was reduced after the microalloying with almost 20%.

## 1. Introduction

The antifriction alloys are a group of alloys with a soft matrix with hard inclusions. The representative alloys have more tin (Sn) and lead (Pb). For this study, an alloy easily found in Romania, with Pb as the main constituent, was studied. A quaternary alloy, YPbSn10, consisting in four elements: Pb, Sb, Sn and Cu, was chosen. An efficient method of improving the properties of antifriction alloys was proved to be microalloying with different elements such as Cd, Ni, As or P [1,2,3,4]. The chemical and physical characteristics determining the lubricant adsorption, chemical affinity towards the conjugated friction surface, dilation, conductivity and thermal fatigue coefficient are linked to the material’s nature. For antifriction alloys we formulated two basic requirements [1,3]: favorable behavior in sliding conditions in a semifluid friction regime and high wear resistance in exploitation. From the mechanical resistance point of view, the constituent phases are prone to be harder, thus the abrasive wear is reduced [4].

Other materials used in tribological applications are particulate reinforced aluminum hybrid metal matrix composites that have increased in automotive applications due to their distinct properties. In the present study, an effort has been made to synthesize Al7075 aluminum alloy with reinforcement of B_4_C and MoS_2_ as lubricant under weight percentages of 4%, 8% and 12% using a stir casting process. A significant improvement of wear resistance and coefficient of friction of aluminum hybrid composites has been achieved owing to inclusion of solid lubricant (MoS_2_) along with hard ceramic reinforcement particles (B_4_C) in the matrix alloy, proved by Shoufa Liu et al. [5].

The material hardness is also influenced by the presence of hard phases. To avoid hard phase agglomeration, a harmful process that might occur during alloy processing, microalloying could represent a good solution [6,7].

In this study, structural and tribological properties of YPbSn10 alloy microalloyed with Ca 2%wt and other samples microalloyed with Mg 2%wt were investigated. The reason for choosing Ca and Mg was that they produce in the structure hard phases that could be uniformly distributed in the structure and they are environmentally friendly elements. The compound CaPb3 has low toxicity and is the hard compound formed when the microalloying is carried out with Ca 2%wt. In the case of Mg microalloying, the hard compound formed is MgPb2 and it presents low toxicity. The novelty of this paper is to use an appropriate amount of Ca and Mg as nontoxic elements to obtain at least a 15% decrease in friction coefficient.

Determining the friction coefficient is essential for the tribological evaluation of antifriction alloys. Such materials, including those described in this paper, have been analyzed by a series of researchers. They determined the friction coefficient of the materials using different types of tribological tests. For example, Nedolini et al. performed pin-on-disk tests on YSn83 [8], CuSn12, CuAl10Fe3 and AlSi12 [9]. Block-on-ring tests were performed by Wang et al. [10] on a ZCuSn10Pb10 alloy, by Amonov et al. on a Sn-based Babbitt metal [11] and Leszczynska-Madej et al. [11] on two different Sn alloys (SnSb12Cu6Pb and SnSb9Cu4). Zeren et al. [12] used for their studies journal bearing test equipment, where the bearings were made from two tin-based bearing alloys (SAE 12 and a Sn–Sb–Cu alloy).

## 2. Materials and Methods

### 2.1. Materials

The alloys produced for the present paper were prepared as follows: two master alloys for YPbSn10 alloy, CuSb50 and PbSb50, Sn and Sb were used. The load was melted at 550 °C and overheated at 600–700 °C. Then, the slag and the coal layer from the metal surface were removed and the rest of the tin was introduced to the mixture. The alloy was maintained for 10–15 min at 500–550 °C, after mixing.

For microalloying the YPbSn10 alloy, adding an extra step to the alloying process was considered. The material was developed as described and the rest of the tin was introduced into the metallic bath. The introduced elements were Ca and Mg, respectively, for the second set of samples. After introducing the tin, the melt was mixed and maintained for 10–15 min at 550–600 °C. The master alloys CuSnCa and CuSnMg were added, respectively, mixed for 1–2 min, then the slag was removed at a temperature of 425–450 °C and poured into a metallic shell.

After introducing the master alloy to the metallic bath, for additional protection of the molten metal bath, an Ar stream was blown into the furnace, which was dried and purified by a CRS purifying cartridge. Ar flow rate: 1–1.5 L/min. Chemical composition of the starting alloy is shown in Table 1.

The chemical composition for the alloy produced in this research was realized by inductive plasma emission spectroscopy (ICP-OES Spectroflame P, Germany) for Sn, Sb, Cu and Pb and with optical emission spectroscopy with continuum current.

For ease of understanding the experiments, the samples were denoted as:
**Alloy****      Sample**YPbSn10Ca0.2      YP1YPbSn10Mg0.2      YP2

For the study, the samples were embedded in Bakelite-type resin and then prepared by grinding with abrasive paper and polished with Lecloth-type cloth soaked with a suspension of α-alumina in water. The attack solution used was C_2_H_6_O + HNO_3_. For the microstructural investigation, an FEI Quanta 250 with high vacuum CBS, ESD and BSE microscope was used. The software used was XTMicroscope server for SEM ELEMENT EDS Analysis Software Suite for EDS.

For XRD analyses, the data acquisition was realized by using a BRUKER D8 ADVANCE diffractometer with the aid of the DIFFRACplusXRD Commender (Bruker AXZ) software, using the Bragg–Brentano diffraction method, coupling Θ-Θ in a vertical configuration. Data processing was realized with the aid of DIFFRAC.EVA VER.5 2019 in the DIFFRAC.SUITE.EVA program package and the ICDD PDF4+2021 database.

### 2.2. Tribological Tests

Tribological tests were carried out with a UMT II BRUKER (former CETR) tribometer. This testing method consists in the use of a cylindrical pin, mounted in the upper part of the tribometer, that applies a constant load on the flat specimen mounted on the lower module that has a translational movement. For these tests, the tribometer was equipped with a dual force sensor model DFH-20 (2 ÷ 200 N range, 25 mN resolution), used to measure the friction force between the upper and lower specimen, as well as to measure and control the normal loading force. In order to maintain a constant loading force during the tests, a suspension system was mounted between the force sensor and the pin holder. The pin has a diameter of 6.35 mm (nominal contact area 31.67 mm^2^) and length of 28 mm and is made of bronze, a material often used in friction couples because of its tribological properties. The materials of the lower specimen are the experimental alloys under study. The tests were performed in dry conditions at room temperature.

The tribometer allows real-time monitoring of the normal load force (*F_z_*), the friction force (*F_f_*) and the friction coefficient (COF).

## 3. Results and Discussions

### 3.1. Structural Characterization

The produced samples’ microstructure analysis results were investigated to observe the influence of Ca on the structure of the alloy. The influence of the mischmetal microalloying on YPbSn10 alloy was present and the beneficial effect of the mischmetal was noticed at 1%wt of mischmetal [13,14].

Figure 1 reveals the compounds formed during microalloying of the base YPbSn10 alloy with Ca 2%wt (YP1 sample).

The compounds formed were subjected to mapping to investigate the elements present in the compound.

Mapping analyses revealed that the compound is formed from Sb, Sn and Cu, suggesting that the compound present here is based on these three elements. In the needle-like compound present in Figure 2, it was observed that copper is present in small quantities.

EDS analysis was performed on the sample and the results are presented in the next section and in Figure 3.

The EDS analyses revealed that the cuboidal compound contains Sb, Sn and Cu, as the mapping shows in Figure 3, and the dark gray needle-like compounds contain more Cu and the light gray mass contains Pb and Ca. Ca is uniformly distributed in the matrix.

The compound containing calcium is present in the highest peak in Figure 4. The XRD patterns confirm the EDS investigation and reveal the Ca distributed evenly in the sample. Other compounds identified in the XRD pattern were Sb and SnSb. The small blue peaks belong to copper stibium compounds. The XRD analyses revealed that the Ca was efficiently microalloyed for the YPbSn10 alloy.

For sample YP2, we investigated the microstructure, and the result is presented in Figure 5.

Figure 5 reveals the cuboidal and needle-like compounds formed during microalloying with Mg 2%wt. The sample YP2 microstructure reveals a series of gray cuboidal compounds evenly distributed in the alloys. Additionally, needle-like compounds with a dark gray color were observed. The bulk alloy is light gray.

Mapping analyses were performed on the cuboidal compound and the results are presented in Figure 6.

For sample YP1, the elements present in the cuboidal compound were Sb and Sn. The needle-like compounds contain Cu and the bulk mass of the alloy contains Pb and MgPb evenly distributed in the bulk alloy. The EDS analyses results identifying the elements in the microalloyed sample are presented in Figure 7.

In this figure, it was observed that the main compounds in the gray cuboidal compounds were Sb and Sn and, in the needle-like shape, more Cu and also Sb and Sn were found.

Figure 8 shows the XRD pattern shows that the red peaks contain Pb and MgPb2 compound and the blue peaks contain mainly Sn and Sb. The MgPb2 compound is a high peak near Pb so the Mg was efficiently introduced in the alloy.

### 3.2. Tribological Tests

In order to determine the friction coefficient (COF) between the three materials and the bronze pin, tribological “pin-on-flat” tests were performed over a length of 5 mm at three different sliding speeds (0.1 mm/s, 0.5 mm/s and 1 mm/s), using three different normal loads (5 N, 10 N and 15 N). Given that the nominal contact area between the pin and the six samples was a circular area determined by the diameter of the pin, the contact pressure corresponding to the three loading forces was 0.16 MPa (5 N), 0.32 MPa (10 N) and 0.48 MPa (15 N).

Figure 9 presents the variation in the friction coefficient for the tests performed on the alloys at 1 mm/s sliding speed and under a normal load of 10 N (0.32 MPa contact pressure). For all the tests, the friction coefficient was relatively constant, indicating a smooth sliding.

The average value of the friction coefficient was determined for all the tests performed and the results are presented in Table 2. It is observed that the relative sliding speed has no significant influence on the average COF. On the other hand, increasing the load leads, in most cases, to a slight increase in the average COF.

The average values of the friction coefficient are graphically analyzed and compared in Figure 10. The analysis results reveal that the lowest values of the friction coefficient were obtained for the sample YP1, with a minimum value of 0.0871. This alloy had a significantly reduced COF compared to the base material (YP3). Comparing the results obtained for samples YP1 and YP2 with those obtained for standard sample YP3, we noticed that the introduction of the alloying materials (Ca and Mg) reduces the friction coefficient considerably, thus improving their tribological behavior.

The good distribution of the hard phase in the alloy mass resulted in decreasing the friction coefficient. It was observed that for a sliding speed of 0.1 m/s, the Ca 2%wt alloyed sample (YP1) had the lowest friction coefficient. It was also observed that for the high sliding speeds employed in the experiments, the sample alloyed with 2% Ca (YP1) had an improved behavior, but also presented the lowest friction coefficient. The Ca added to this alloy was more beneficial than Mg. The CaPb compounds were well distributed in the alloy and the tribological results revealed a decrease with almost 20% of the friction coefficient in comparison with the base alloy YPbSn10. The improvement in the structure, by adding calcium or magnesium is proved by the improvement obtained in the tribological properties. The decrease in the friction coefficient value is a good measure for tribological properties.

## 4. Conclusions

Two novel compositions of YPBSn10 alloy, microalloyed with 2%wt Ca or 2%wt Mg, were produced.

The alloy characterization was performed from a structural and a tribological point of view. The investigated structure was homogeneous, and the hard phase was well distributed in the alloy matrix. The XRD analyses underlined the compounds formed during alloy production.

Based on the tribological tests, it was determined that microalloying the YPBSn10 alloy with 2%wt Ca and 2%wt Mg, respectively, leads to an improved friction coefficient. However, by comparing the two alloys, the one containing Ca as the microalloying element had an improved behavior during tribological tests, exhibiting the lowest friction coefficient for different testing conditions. Thus, for an improved sliding behavior it is recommended to use the Ca 2%wt for microalloying of the YPbSn10 alloy for different tribological applications.

## Figures and Tables

**Figure 1 materials-15-03289-f001:**
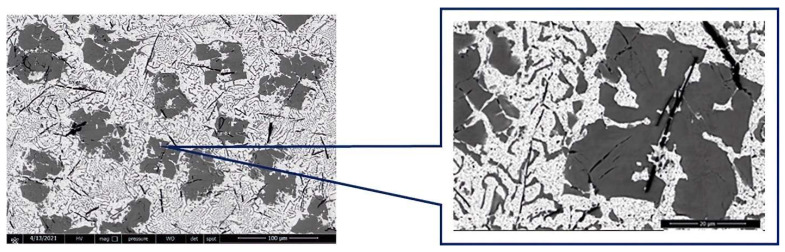
SEM image of YP1 sample at 100 µm and 20 µm.

**Figure 2 materials-15-03289-f002:**
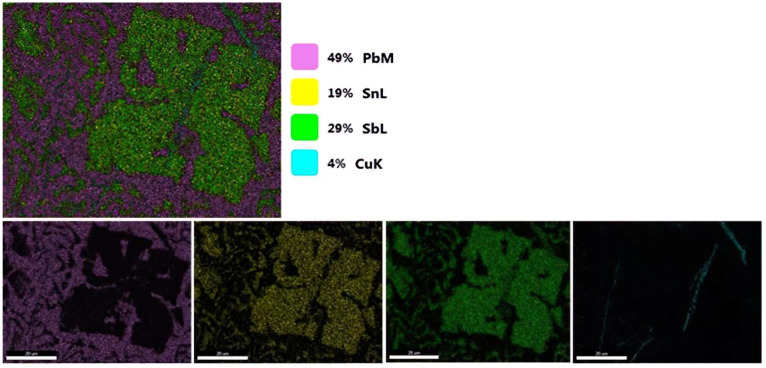
Mapping analyses of the compounds formed during Ca microalloying.

**Figure 3 materials-15-03289-f003:**
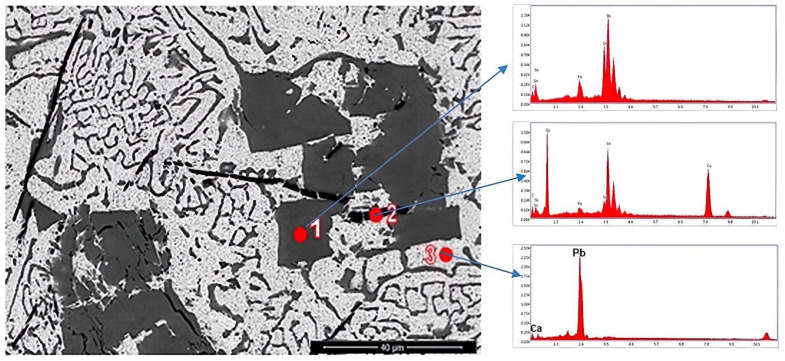
EDS analysis in points 1, 2 and 3 for the sample YP1.

**Figure 4 materials-15-03289-f004:**
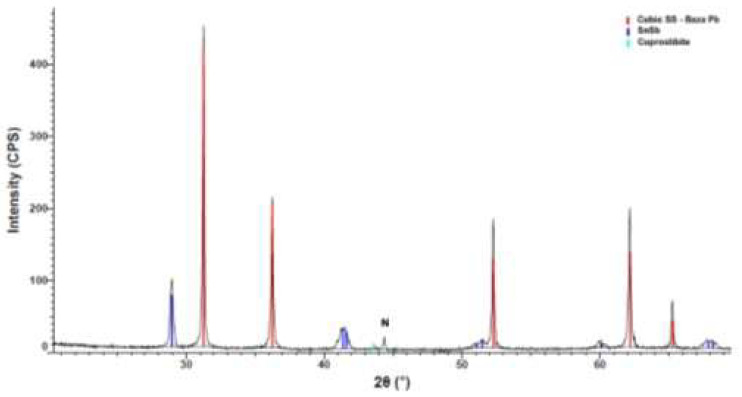
XRD pattern of the YP1 sample.

**Figure 5 materials-15-03289-f005:**
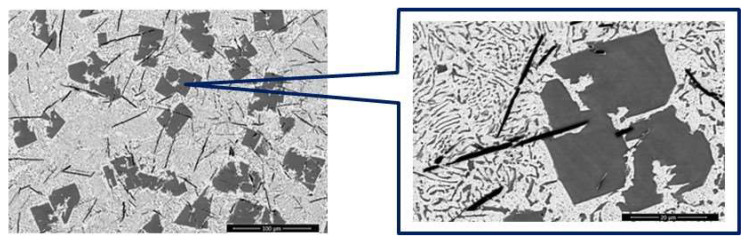
SEM image of YP2 sample at 100 µm and 20 µm.

**Figure 6 materials-15-03289-f006:**
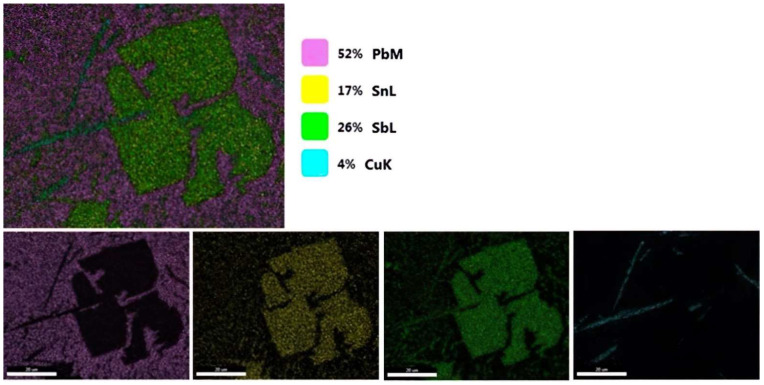
Mapping for the cuboidal compound found in sample YP2.

**Figure 7 materials-15-03289-f007:**
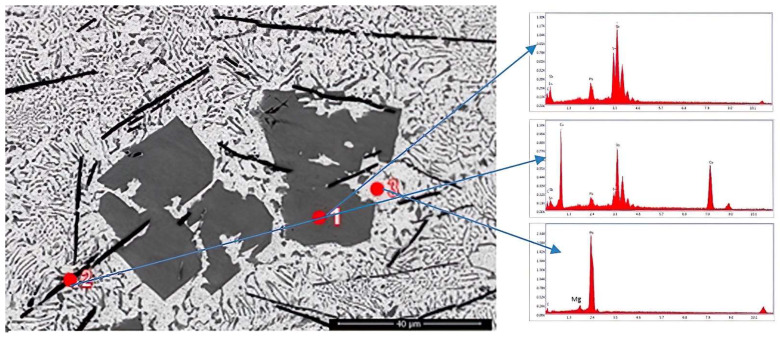
EDS analyses results for YP2 sample.

**Figure 8 materials-15-03289-f008:**
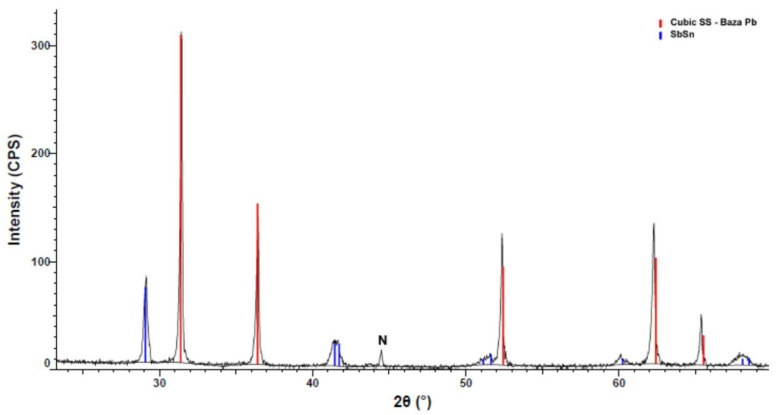
XRD pattern for the sample YP2.

**Figure 9 materials-15-03289-f009:**
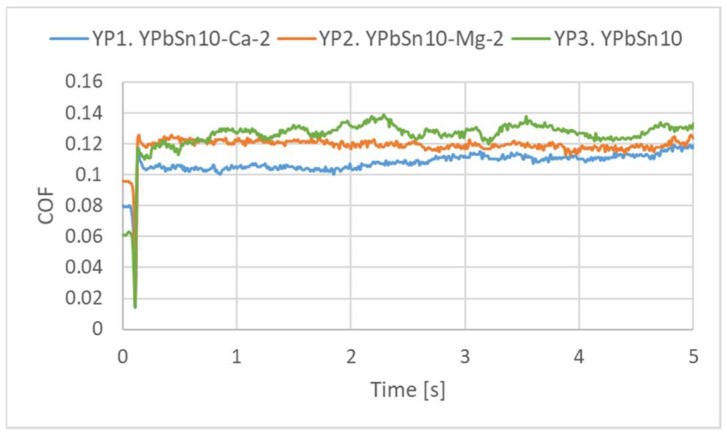
Variation in the COF during the 10 N, 1 mm/s tests.

**Figure 10 materials-15-03289-f010:**
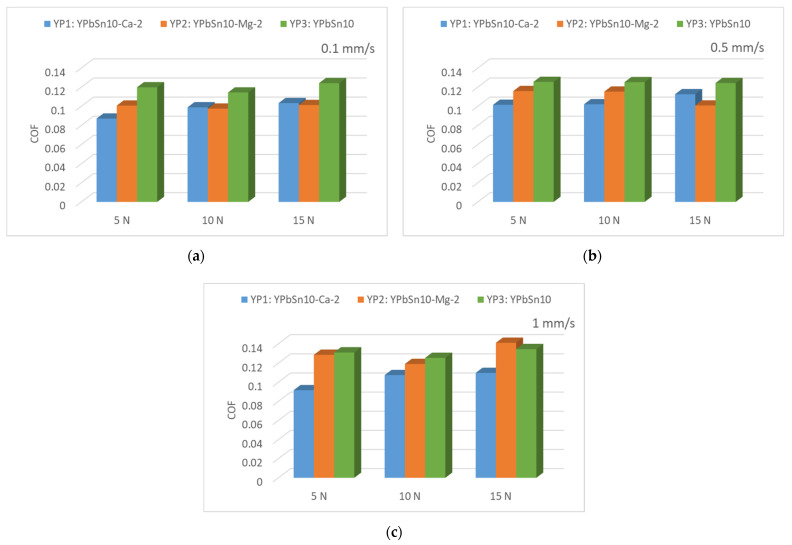
Average values of the friction coefficient: (**a**) Comparison for the 0.1 mm/s sliding speed; (**b**) comparison for the 0.5 mm/s sliding speed; (**c**) comparison for the 1 mm/s sliding speed.

**Table 1 materials-15-03289-t001:** Chemical composition of alloys produced.

Sample	Pb	Cu	Sb	Sn	Zn	Fe	Al	Ni	Mo	Ca	Mg	Others *
YP1	bal	1.03	13.5	8.7	<0.02	<0.02	<0.02	<0.02	<0.02	0.23	-	<0.01
YP2	bal	1.19	13.9	7.1	<0.02	<0.02	<0.02	<0.02	<0.02	-	0.21	<0.01

* Others: impurities that could be found in this alloy as S, Mo, Al, As, aso.

**Table 2 materials-15-03289-t002:** Average value of the friction coefficient.

Sample	Loading Force *F_z_*	Relative Sliding Speed		
		0.1 mm/s	0.5 mm/s	1 mm/s
YP1: YPbSn10-Ca-2	5 N	0.0871	0.1015	0.0916
	10 N	0.0989	0.1021	0.1074
	15 N	0.1033	0.1126	0.1097
YP2: YPbSn10-Mg-2	5 N	0.1007	0.1159	0.1286
	10 N	0.0975	0.1153	0.1190
	15 N	0.1013	0.1008	0.1410
YP3: YPbSn10	5 N	0.1198	0.1255	0.1310
	10 N	0.1144	0.1252	0.1254
	15 N	0.1242	0.1243	0.1344

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
