# Peer review of "The Effect of Ca and Mg on the Microstructure and Tribological Properties of YPbSn10 Antifriction Alloy"

_materials, 2022, doi:10.3390/ma15093289_

Round 1

Reviewer 1 Report

In this research study the authors obtained two novel compositions of YPBSn10 alloy, microalloyed with 2%wt Ca and respectively 2%w tMg, in which they characterized the alloy obtained from structural and tribological point of view.

Also, the authors succeeded to show that the investigated structure was homogeneous, and the hard phase was well distributed in the alloy matrix.

Also, they proved that the microalloying technique have good results in producing alloys with homogeneous composition, with a good distribution of the hard phase, in which the alloys were produced in a furnace and then sample collected and investigated, then the structural properties were investigated using SEM technique with EDS analyses and XRD to identify the compounds formed during alloying.

The authors also studied the tribological properties in which they were investigated to see the improvement obtained in this area. The results revealed a homogeneous composition for both samples, alloyed with Ca respectively with Mg and the friction coefficient improvement was evident after the test performed, which makes the studied research area is so unique and novel.

Author Response

Dear Reviewer 1,

Thank you for your kind comments. We carefully checked the manuscript for English errors and revised everything.

Best regards,

The authors

Reviewer 2 Report

1) Kindly please enhance the language standard

2) Kindly please maintain the journal format

3) Only limited reference works are available. You may include many references. You may include the following also in the perspectives alloys elements.

Shoufa Liu, Yinwei Wang, et al, “Effect of B4C and MOS2 reinforcement on micro structure and wear properties of aluminum hybrid composite for automotive applications,” Composites Part B: Engineering, 176, 107329, 2019.

4) Figure 2 , 4 quality may be improved

5)  What is main interpretation of XRD analysis?

6)  What is the specific application of proposed material? Will it satisfy he requirements for the application.

7) How many trials have been made for every experimental trial?

8) Kindly refine the conclusion section

9) Kindly please include the title for Reference section

Author Response

Dear Reviewer 2

Thank you for your comments. You help us to improve our manuscript.

We checked the manuscript for English. We checked the format and keep it for the manuscript. WE included the indicated reference.

We increased the quality for figure 2 and 4.

We added more details for XRD interpretation.

The application of these alloys are in tribology field.

We performed several trials for each experiment.

WE refined the conclusion section

Brest regards,

The authors

Reviewer 3 Report

In this paper, effect of Ca and Mg on the microstructure and tribological properties of YPbSn10 antifriction alloy were investigated. However, the discussions and analyses appear incomplete. There are several major points that needs to be addressed.

  1. The language of the manuscript must be thoroughly revised. A large number of sentences are badly constructed and some grammatical errors are present. For instance: “…, the structure was improved be microalloying technique”, “The load was melted at ??? and overheated at 600 - 700°C.”
  2. The authors mentioned “the friction coefficient improvement was evident after the test performed” in the abstract. However, friction coefficient improvement usually means decreasing tribological properties.
  3. Novelty of the research is not clear.
  4. The abbreviated forms of the phrase should be not used for the first time.
  5. The authors should provide clear images in the present paper. XRD patterns should be indexed in Figure 5 and Figure 9.
  6. EDS results in Figure 4 and Figure 8 should be provided.
  7. More discussion on the effect mechanism of the microstructure on tribological properties should be made.
  8. The forms of references should be consistent.

Author Response

Dear Reviewer 3,

Thank you for your useful comments. You helped us to improve our paper.

We carefully revised the English language and corrected all the errors.

We defined that the friction coefficient decreased with a significant percentage and the wear properties were improved.

We studied the tribological properties in and we investigated to see the improvement obtained in this area. The results revealed a homogeneous composition for both samples, alloyed with Ca respectively with Mg and the friction coefficient improvement was evident after the test performed, which makes the studied research area novel.

We revised the abbreviation along the paper.

We made the images clearer and the pattern was marked with different colors.

EDS data are on the graph presented

We added more discussion for the microstructure on tribological properties.

We verified all the references.

Thank you,

The authors

Reviewer 4 Report

This paper compares the microstructure and tribological behavior of a commercial Pb-based alloy prepared as two micro-alloyed variants by well-known methods against the original material. Results show the existence of some hard phases that favorably reduce the coefficient of friction.

In my opinion, the paper is not clear about its novelty and significance because although the authors declare that the produced alloys are of novel compositions, they do not compare them against other state-of-art alloys to demonstrate better tribological properties. A formal discussion must be addressed to evaluate the novelty and then provide the readers with an objective point of view to demonstrate the advantages of the obtained alloys.

Some other minor issues are:

  1. The abstract should be improved. It does not include representative numerical results, such as the friction coefficient or the identified compounds formed during alloying.
  2. In the Materials section, there is a missing value or text (marked as ???). The identification of the sample’s nomenclature (YP1, YP2) should be expressed in the text, and the YP3 alloy (original commercial piece) is not mentioned here.
  3. It is not necessary to include the figure of the tribometer. It is a well-known technique and equipment used for tribological studies.
  4. There are various minor grammar errors and typos in the whole text, and the punctuation should be revised.

Author Response

Dear Reviewer 4,

Thank you for your comments helping us to improve our paper.

We revised the English language and punctuation for our paper.

We corrected the mistake with ??? and checked all the manuscript.

We revised the abstract.

Thank you,

The authors

Round 2

Reviewer 2 Report

The manuscript may be accepted in its present form.

Author Response

Dear Reviewer,

Thank you for your appreciation.

Best regards,

The authors

Reviewer 3 Report

The authors have revised the manuscript according to the review comments. However, the authors must modify these before the publication may be accepted.

  1. The chemical formula “B4C and MoS2” in the manuscript should be revised into “B4C and MoS2”.
  2. The figures in the manuscript are still not clear.

Author Response

Dear Reviewer,

Thank you for your comments. The formulae have been changed and the quality of the figures were improved. Also we checked again the manuscript for English to remove all the minor errors.

Thank you and best regards,

Authors

Reviewer 4 Report

I still recommend that the authors revise the novelty and significance of this research. Please include a deep discussion and compare the results with state-of-the-art. 

Also, I noticed that the authors have decided to include figure 1, so please explain in the paper why this is relevant or what is the difference between your tribometer and other commercial equipment. 

The quality of all figures must be improved. 

Minor writing mistakes are still present. 

Author Response

Dear reviewer,

Thank you for your comments. We revised the novelty, we added a comparison with the state of the art. We improved the picture quality and we checked again the manuscript for English errors. Also we removed figure 1.

Best regards,

The authors
